# Acute Kidney Injury in Cancer Immunotherapy Recipients

**DOI:** 10.3390/cells11243991

**Published:** 2022-12-10

**Authors:** Adrien Joseph, Antoine Lafarge, Elie Azoulay, Lara Zafrani

**Affiliations:** Medical Intensive Care Unit, Saint-Louis Teaching Hospital, Public Assistance Hospitals of Paris, 75010 Paris, France

**Keywords:** immunotherapy, acute kidney injury, immune checkpoint inhibitors, chimeric antigen receptor T cell therapy, cancer

## Abstract

Cancer immunotherapy has now entered clinical practice and has reshaped the standard of care for many cancer patients. With these new strategies, specific toxicities have emerged, and renal side effects have been described. In this review, we will describe the causes of acute kidney injury in CAR T cell, immune checkpoint inhibitors and other cancer immuno-therapy recipients. CAR T cell therapy and bispecific T cell engaging antibodies can lead to acute kidney injury as a consequence of cytokine release syndrome, tumor lysis syndrome, sepsis or specific CAR T cell infiltration. Immune checkpoint blockade most often results in acute tubular interstitial nephritis, but glomerular diseases have also been described. Although the pathophysiology remains mostly elusive, we will describe the mechanisms of renal damage in these contexts, its prognosis and treatment. As the place of immunotherapy in the anti-cancer armamentarium is exponentially increasing, close collaboration between nephrologists and oncologists is of utmost importance to provide the best standard of care for these patients.

## 1. Introduction

Immunosurveillance against cancer has been acknowledged for more than a century, when Wilhelm Busch and Friedrich Fehleisen described the association between immune status and cancer and Sir William Colley inoculated cancer patients with extracts of inactivated *Streptococcus pyogenes* and *Serratia marcescens* to induce tumor regression [1]. A few decades later, Paul Ehrlich hypothesized the concept of immune surveillance against cancer [2], but the lack of basic knowledge precluded the translation into clinical practice, and the advent of radiotherapy and cytotoxic chemotherapy displaced immunotherapy as a pillar of cancer treatment.

The idea to enhance the immune system to treat neoplastic disease resurfaced with the advent of bioengineering, knockout mouse models and molecular and biomedical advances which helped to decipher tumor-specific immune responses and put the light on the specific role of T cells in antitumor immunity [3].

These basic discoveries led cancer immunotherapy to revolutionize the field of oncology, becoming a first-line treatment in an unprecedentedly wide range of indications. Immunotherapy acts either by unleashing the intrinsic power of the immune system (immune checkpoint inhibitors, bispecific antibodies) or through the bioengineered enhancement of immune cells (chimeric antigen receptor (CAR) T cells) (Table 1).

The flip side of this paradigm shift was a new spectrum of side effects related to their specific mechanisms of action. Cancer patients are known to be at high risk of acute kidney injury, resulting from various tumor-specific and treatment-specific insults [4,5,6]. Side effects of cancer immunotherapy are no exception to the rule, with specific renal effects that physicians must recognize in order to provide the best standard of care and improve these patients’ prognosis.

In this narrative review, we will describe the causes of acute kidney injury in recipients of cancer immunotherapy (Figure 1) (Table 2). We will only focus on FDA or EMA-approved cancer immunotherapies (Table 1), but readers must keep in mind that immunotherapy is a rapidly evolving field, and newly identified targets and novel treatment combinations will only expand the role of immunotherapy in the treatment of cancer and its kidney-associated side effects in the decades to come.

## 2. Acute Kidney Injury in CAR T Cells Recipients

Chimeric antigen receptor (CAR) T cells are an adoptive cell therapy [7] that consist of autologous T cells transfected with a lentiviral vector encoding for a variable light region of an antibody, a spacer structure anchored to the cellular membrane and a region of the TCR CD3ζ chain with one or more co-stimulatory receptors. This unique architecture is able to recognize the antigen in the absence of major histocompatibility complex (MHC) presentation as opposed to native T cell receptors [8].

CAR T cells’ variable regions are directed against a tumor target of interest, most often CD19 for B-lineage lymphomas and leukemias and more recently B cell maturation antigen (BCMA) for multiple myeloma [9,10,11].

CAR T cells were first approved in 2017 [12] for the treatment of acute lymphoblastic leukemia and have now entered an era of fast-paced and innovative research [8], which is accompanied by a tremendous number of clinical trials for a broad array of hematological and solid malignancies [13,14].

Patients first undergo cytapheresis followed by a lymphodepleting chemotherapy (often a combination of cyclophosphamide and fludarabine) to create an environment conducive to the infusion of autologous engineered CAR T cells.

The infusion of CAR T cells has been associated with specific toxicities [15]: cytokine release syndrome (CRS) and immune effector cell associated neurotoxicity syndrome (ICANS), that result from T cell expansion. Non-specific toxicities from on-target effects can also occur, such as sepsis and tumor lysis syndrome. These complications put CAR T cells recipients at high risk of AKI.

The incidence of AKI in patients treated with CAR T cells varies according to indication (lymphoblastic leukemia, lymphoma or myeloma), patients’ characteristics (age, baseline renal function), tumor burden and occurrence of CAR T cells complications. The overall incidence rate is estimated around 17%, with 2.9% requiring renal replacement therapy [16]. The prognostic impact of AKI is also important, with a mortality risk up to 67% at 60 days compared to 10% in CAR T cell recipients without AKI [17].

The mechanisms responsible for AKI in the context of CAR T cell therapy are detailed hereafter.

### 2.1. Cytokine Release Syndrome (CRS)

As a result of CAR T cells expansion, approximately three-quarters of patients will experience some degree of cytokine release syndrome [16,18]. CRS is a potentially life-threatening complication in which activated immune cells (CAR T and mostly recipients macrophages) [19] secrete high levels of inflammatory cytokines including IL-1RA, IL-2, IFNγ, IL-6, soluble IL-6R, IL-8 and GM-CSF. After a median of 5 days, patients usually develop fever with varying degree of hypotension, hypoxia and other organ failures according to the severity of the disease [15,20,21]. Rarely, CRS can evolve into fulminant hemophagocytic lymphohistiocytosis and multiorgan failure, requiring organ support and admission in an intensive care unit [22,23].

High-grade CRS is associated with a five to ten times greater risk of developing AKI [17,24,25]. Acute kidney injury in this context results from inflammatory cytokine-driven vasodilation, vascular leak and subsequent decreased renal perfusion [26], potentially leading to acute tubular necrosis if persistent. Nevertheless, a lack of correlation between inflammatory markers such as interleukins or ferritin and occurrence of AKI [25] highlights the diversity of molecular mechanisms leading to AKI in the context of CRS.

CRS has also been occasionally associated with collapsing glomerulopathy [27], which in this context may have resulted from cytokine-induced podocyte and endothelial injury [28,29], as seen in patients with hemophagocytic lymphohistiocytosis [30,31]. The role of APOL-1 mutations which, in this context, may contribute to the pathogenesis of collapsing glomerulopathy, creating a double-hit phenomenon, has never been explored.

CRS treatment consists of non-specific intensive care measures, such as intravenous fluids, vasopressors and occasionally ventilatory support, along with specific cytokine-blocking therapies, namely IL-6 (siltuximab), IL-6R (tocilizumab) or IL-1 (anakinra) blockers and corticosteroids [15,20,22]. Guidelines have been published, with consensus grading and therapeutic strategy, but optimal care remains to be determined and mostly relies on expert opinion and multidisciplinary discussions (Table 3). Early and effective treatment is likely to decrease the risk of CRS-associated AKI, but substantiating evidence supporting this assertion is lacking.

### 2.2. Tumor Lysis Syndrome (TLS)

After CAR T cell-mediated destruction of target cells, high amounts of intracellular components such as potassium, phosphate and deoxyribonucleic acid [32] are released into the circulation, leading to hyperkalemia, hyperuricemia (from DNA catabolism), hyperphosphatemia and subsequent hypocalcemia. Uric acid and calcium phosphate deposition can lead to acute tubular precipitation and injury, but more recent data also shed light on the role of extracellular histones and endothelial cell activation in the pathogenesis of TLS [33].

TLS has been rarely (<5%) reported in early trials of CAR T cell therapy [34,35,36], but it can be severe and even lethal [18]. More efficient strategies and treatment of patients with higher disease burden will likely translate into increased incidence of TLS after CAR T cell therapy.

Electrolyte abnormalities have also been noted outside TLS. Most notably, hypophosphatemia and hyponatremia can occur after a median 5–6 days following CAR T cell administration [17], presumably resulting from perfusion fluids, inappropriate antidiuretic hormone secretion and undernutrition.

### 2.3. CAR T Cells Infiltration

Anecdotally, CAR T cells have been reported to infiltrate renal parenchyma in a kidney transplant recipient, leading to acute cellular rejection [37]. The reason for renal CAR T cells migration into the renal parenchyma is unknown, as their antigenic target, CD19, is scarcely distributed within this tissue [38].

Nevertheless, as coagulation disorders and thrombocytopenia [39,40] often preclude renal biopsy in the acute phase after CAR T cell therapy, one can hypothesize that CAR T cells renal infiltration is an underestimated mechanism of AKI in these patients.

### 2.4. Sepsis

Sepsis in the context of CD19 CAR T cells therapy can result from on-target and off-tumor effects, leading to B cell aplasia. Conditioning regimen and neutropenia also put CAR T cells recipients at high risk of infection [41]. During the first 28 days after CAR T cells administration, bacterial infections are most frequent, and 20–40% of CAR T cell recipients will experience infectious complications [32]. During the first year after CAR T cells infusion, the cumulative incidence of overall, bacterial, viral, and fungal infections were 63%, 57%, 45%, and 4%, respectively [42], and infectious complications have been reported as the main cause of non-relapse complication in CAR T cells recipients [43].

Mechanisms underlying sepsis-induced AKI are diverse and involve local inflammation, metabolic reprogramming [44] and microvascular endothelial dysfunction [45].

Due to its high prevalence in CAR-T recipients and the difficult differential diagnosis between CRS and sepsis, empirical broad-spectrum antibiotics are advocated for severe patients with circulatory failure and acute kidney injury in the first month after CAR T cell administration.

Infection monitoring and antimicrobial prophylaxis (including herpesvirus and *pneumocystis jirovecii* prevention) are warranted after CAR T cells infusion. Antifungal prophylaxis, vaccinations and prophylactic IgG infusions are also discussed, depending on previous and concomitant treatments and immune reconstitution [46].

### 2.5. Tumor Progression

Finally, AKI can be associated with tumor progression in the context of CAR T cells failure. Various mechanisms, such as lymphoma interstitial infiltration, myeloma cast nephropathy or renal obstruction can be responsible for AKI during hematological malignancies, and these fall beyond the scope of this review [5].

## 3. Acute Kidney Injury in Immune Checkpoint Inhibitors Recipients

Currently approved immune checkpoint inhibitors (ICI) are monoclonal antibodies that act by reversing tumor escape caused by two negative regulators of tumor immunity: cytotoxic T-lymphocyte antigen 4 (CTLA-4) and programmed cell death 1 (PD-1) or its ligand, programmed cell death ligand 1 (PD-L1). These receptors are expressed on T lymphocytes, other immune cells, and tumor cells [47]. Their blockade prevents T cell co-stimulation by outcompeting CD28-B7 association and T cell effector functions by blocking PD1-PD-L1 engagement.

Since the first approval of ipilimumab for melanoma in 2011, six anti-PD-1/PD-L1 antibodies and one anti-CTLA-4 antibody have been approved, and ICI have reshaped the standard of care in solid and hematological malignancies, yielding unprecedented results in an unparalleled wide range of indications [48], as recognized by the 2018 Nobel Prize award for Medicine and Physiology. As for 2019, an estimated 40% of patients with cancer in the United States were eligible for treatment with an ICI [49].

Overall, the safety profile of ICI is favorable compared with conventional chemotherapy [50]. However, overactivation of the immune system can lead to immune-related adverse events (irAEs) that differ markedly from classical chemotherapy toxicities. These irAEs can affect any organ and grossly mimic the spectrum of autoimmune organ diseases, often without significant serological findings [51].

The proportion of patients affected by irAEs depends on the therapeutic target, tumor type [52] and recipient characteristics, such as younger age [53], microbiota composition [54] and history of autoimmune disease [55,56]. It can reach 90% for all grade and 40% for grade 3–4 adverse events in patients treated with anti-PD1 / anti-CTLA4 combination [51].

If the skin, gastrointestinal tract and liver are most frequently affected, the kidney is no exception to the rule and can be the target of overactivation of the immune system. Kidney involvement is the most delayed irAE, with a median time from ICI initiation to onset of AKI of 14 (IQR 6–37) weeks and represents a rare complication, affecting 1% (single agent) to 5% (combination therapy) of treated patients. A longer delay of AIN onset, from 3 to 12 months, was reported in patients receiving anti-PD-1 and/or anti-PD-L1 anti-bodies [53,57,58,59,60].

### 3.1. Acute Tubular Interstitial Nephritis

Acute tubular interstitial nephritis is the most commonly reported renal lesion found in published case series of ICI treated patients [57,58,59,61,62].

The pathophysiology of ICI-associated acute tubular interstitial nephritis differs from that of other drug-related acute interstitial nephritis, and it likely relies on the loss of self-tolerance versus self-renal antigens, as opposed to delayed hypersensitivity reaction. Other hypotheses include off-target effects on tubular cells overexpressing PD-L1 [63] and the generation of a proinflammatory milieu [64].

Proton-pump inhibitors and non-steroidal anti-inflammatory drugs are encountered in an unexpectedly high proportion of patients with ICI-associated acute tubular interstitial nephritis [53,57,58], leading some authors to hypothesize a role of autoreactive T cells reacting to these nephrotoxic agents unleashed by ICI treatment [61]. Fluindione, an antivitamine K antagonist known to induce acute drug-induced interstitial nephritis, has also been associated with acute tubular nephritis occurrence in the context of ICI treatment [65] and reinforces the hypothesis of re-activation of drug-specific T cells as the main mechanism of acute tubular interstitial nephritis.

Its clinical presentation includes mild, non-selective proteinuria (around 0.5 g/g creatininuria), leukocyturia in approximately half of the patients and renal dysfunction that requires renal replacement therapy in up to 10% of patients. Notably, most patients do not present with eosinophilia, and approximately half of patients have multiorgan irAEs, pre-existing or occurring with acute tubular interstitial nephritis [57,58].

Risk factors for ICI-associated AKI include younger age [53], lower baseline glomerular filtration rate, anti-CTLA4 and most importantly anti-CTLA4/anti-PD1 combination therapy [57].

Baseline renal impairment from a non-immune origin does not contra-indicate ICI therapy [66], but, as expected, renal transplant recipients are at high risk of transplanted kidney rejection under ICI treatment, especially with anti-PD1 treatment [67,68,69,70]. Switching from calcineurin to mTOR inhibitors is an interesting strategy in this context with the objective of uncoupling anti-PD-1 therapy toxicity and efficacy, as mTOR inhibitors have been shown to induce Treg expansion [71,72], have an independent antitumor effect [73] and are associated with fewer incidents of cancer occurrence compared to calcineurin inhibitors [74,75,76]. However, this approach, only reported in case reports [77,78], has yet to be validated.

Kidney biopsy usually shows CD3+ lymphocytic infiltrates with varying degrees of plasma cells and eosinophils, and some patients display granulomatous features [58].

Immunofluorescence typically yields background staining for C3 along vessel walls without tubular basement membrane or glomerular staining.

The question of whether to perform kidney biopsy when facing creatinine elevation during the course of ICI treatment is a matter of debate [79,80,81]. Acute kidney injury in the context of cancer is common and results from various mechanisms, including other nephrotoxic drugs, crystalline nephropathy, ischemic tubular injury, paraneoplastic kidney damage and postrenal AKI.

A tubulointerstitial profile with low-grade proteinuria and increased ^18^F-flourodeoxyglucose uptake in the renal cortex of patients undergoing ICI treatment for more than 2 weeks in positron-emission tomography imaging [82] can corroborate the decision to forego the need for kidney biopsy and proceed with immunosuppresive therapy, as advocated in the ASCO guidelines [66], even though this attitude is not supported by sound scientific evidence and should not delay corticosteroids initiation. On the contrary, the presence of glomerular proteinuria, hematuria, thrombotic microangiopathy features or AKI refractory to steroids and other immunosuppressant agents must prompt histologic evaluation. Unnecessarily withholding ICI therapy because of immune-associated tubular interstitial nephritis overdiagnosis can compromise patients’ oncologic outcomes, and plausible alternative diagnosis should also encourage physicians to perform kidney biopsy.

Importantly, in the largest published cohort, no histologic feature, including the presence/severity of granuloma, tissue eosinophilia, interstitial fibrosis, or glomerulosclerosis, was associated with kidney recovery, and the prognoses of patients were similar irrespective of kidney biopsy [57].

First-line treatment of irAEs mostly relies on corticosteroids and ICI discontinuation in severe cases [66], but targeted therapies are increasingly used to treat refractory or severe organ damage [83]. The American Society of Oncology (ASCO) guidelines recommend temporarily withholding ICI for grade 2 nephrotoxicity (creatinine 2–3× above baseline) and discontinuation of treatment for grade 3 or 4 (creatinine > 3× baseline), and initiation of corticosteroids if symptoms persist for more than 1 week in grade 2 and immediately in grade 3 or 4 toxicity [66]. Specifically, acute tubular interstitial nephritis is often treated with 1 mg/kg/d prednisone equivalence units, and some severe patients are treated with pulse intravenous methylprednisolone. No tapering strategy has been prospectively evaluated in this indication, but a minimum of 4 weeks is advocated [66] and treatment is often prolonged for a total of 3 months [57] (Table 4). Additional immunosuppressive drugs are used in less than 10% of patients and include mycophenolate mofetil, azathioprine, rituximab or cyclophosphamide [57]. The paucity of data precludes any definitive conclusion regarding second-line immunosuppressive strategy in this context.

The clinical course of ICI-associated acute tubular interstitial nephritis differs from other drug-induced acute interstitial nephritis in that patients have a slower response to corticosteroids and slower recovery. An estimated 40% of patients with acute tubular interstitial nephritis have complete renal recovery, whereas incomplete or no recovery occur in 45% and 15% of patients. Less than 10% require renal replacement therapy and among them, half will achieve dialysis independence. Renal recovery has been associated with concomitant exposure to a tubular interstitial nephritis-causing medication and treatment with steroids, whereas simultaneous extrarenal irAEs were associated with poorer recovery [57].

The readministration of ICI after irAE resolution relies on a thorough evaluation of the risk–benefit ratio. In general, ASCO guidelines consider rechallenge if irAEs revert to grade 1 (serum creatinine < 2× baseline value) but warrant caution in early onset irAE and advocate permanent discontinuation in extra-endocrinologic grade 4 irAEs [66]. Several case series report on the readministration of ICI after severe irAEs with an acceptable safety profile under close monitoring [84,85].

In the specific context of acute tubular interstitial nephritis, rechallenge with an ICI appears to be at higher risk of recurrence, compared with colitis or hypophysitis [85], and results in approximately one-fourth of AKI recurrence [57].

It is unclear whether acute interstitial nephritis is associated with clinical response to ICI therapy [86], as shown for other immune-related adverse events [87]. The occurrence of acute tubular interstitial nephritis does not seem to be associated with poorer outcomes [53,88]. Nevertheless, failure to achieve kidney recovery is associated with worse overall survival [57].

### 3.2. Others

In addition to tubulointerstitial nephritis, a wide range of kidney lesions have been associated with ICI therapy. Renal vasculitis, pauci-immune glomerulonephritis ANCA +/− [89], minimal change disease, acute tubular injury, anti-glomerular basement membrane disease and C3 glomerulonephritis [57,58] have also been described.

Thrombotic microangiopathy cases have been reported and are often limited to renal parenchyma, without systemic hemolytic anemia and schistocytosis, and they may carry a poorer response to corticosteroids [58].

In addition to acute kidney injury, PD1 blockade has been associated with an increased risk of hypocalcemia [60], which warrants routine monitoring.

With the expansion of ICI indications and the advent of novel therapeutic targets [90], the number of treated patients and irAEs is expected to continue to rise. For example, a recently published phase 2–3 study combining anti-LAG3 and anti-PD1 antibodies in melanoma patients reported 7/355 (2%) acute renal dysfunction [91], whereas this adverse events was not described in the original studies describing anti-CTLA4/anti-PD1 combination [92].

## 4. Acute Kidney Injury in Other Cancer Immunotherapy Recipients

### 4.1. Bispecific Antibodies

Since their coming of age in the 2010s, monoclonal antibodies have relied on antibody-dependent cell-mediated cytotoxicity and complement-dependent cytotoxicity to exert their antitumor effect [93]. The idea to exploit the potential of the host immune system later converted the technological advances of bioengineering into T-cell engaging bispecific antibodies [94], with the approval of blinatumomab, an anti-CD3/anti-CD19 antibody in 2015 for acute lymphoblastic leukemia.

Bispecific T cell engaging therapy consists of two single-chain variable fragments directed against both the TCR complex and a tumor-associated antigen. CD19 is currently the most widely used target (Blinatumomab), but bispecific antibodies redirecting T cells against CD20 (Mosunetuzumab) have recently been approved for follicular lymphoma and new antibodies targeting BCMA for multiple myeloma are likely to enter clinical practice in the near future [95].

Bispecific T cell engaging antibodies are thus not only capable of redirecting the cytotoxic potential of T cell toward a given target, but also, in contrast to ICIs, do not require MHC T cell response of the host [96].

Akin to CAR T cells, CD3/CD19 bispecific T cell engaging antibodies can cause cytokine release syndrome, neurological specific toxicities, tumor lysis syndrome and sepsis. Around 14% of adult patients receiving blinatumomab will experience CRS (5% grade ≥ 3), but the incidence of severe AKI (1–2%) does not seem to differ from conventional chemotherapy treatment. Sepsis appears to be almost 10 times less frequent with blinatumomab compared to standard chemotherapy [97,98], even though blinatumomab, as a result of its on-target effects, cause profound and durable B cell aplasia and hypogammaglobulinemia [99,100]. On the other hand, tumor lysis syndrome and its associated electrolyte abnormalities tended to occur more frequently in blinatumomab-treated patients [97]. Acute interstitial nephritis with T cell infiltration has not been described in the context of CD3/CD19 bispecific T cell engaging antibodies, as opposed to CAR T cells, presumably as a result of less broad immune cell activation.

### 4.2. BCG Therapy

Intravesical *Bacille Calmette-Guerin* (BCG) therapy has been used since 1976 for non-muscle invasive bladder cancer and represents one of the first immunotherapeutic intervention in the modern era of cancer treatment. Although its precise mechanism remains putative, it is thought to rely on a cross-presentation mechanism involving cellular immunity and the chemokine-mediated recruitment of immune cells [101,102,103]. Common side effects include transient cystitis, dysuria and low-grade pyrexia [104], but more severe BCGitis occurs in less than 5% of patients [105], particularly in the case of traumatic instrumentation and vascular breach.

Acute kidney injury complicates around 40% of BCGitis [106] and usually presents with acute interstitial nephritis, with or without granuloma [107,108]. *Bacillus Calmette et Guérin* can disseminate and cause renal infection, but it is assumed that BCGitis and granulomas are more often caused by type 4 hypersensitivity reactions, as evidenced by the lack of mycobacterial isolation in urine culture or kidney biopsy [109]. Anecdotal reports of mesangial glomerulonephritis [107,110], membranous nephropathy and hemolytic uremic syndrome have also been published [111,112,113], possibly in relation with an autoimmune response to BCG.

By analogy with *M. bovis* infection, a 6–9 months anti-tuberculous tritherapy seems justified, in association with corticosteroids [104,105].

### 4.3. Others

Other approved cancer immunotherapies, such as oncolytic viruses (T-VEC) for melanoma [114] or sipuleucel-T for prostate cancer [115] are less frequently prescribed and have not been associated with specific renal side effects.

## 5. Conclusions

Cancer immunotherapy has reshaped cancer therapy and led to the description of new toxicities, including renal toxicities. Cytokine release syndrome and sepsis are the main mechanisms for CAR T cell therapy and bispecific T cell engaging antibodies acute kidney injury and require a rapid evaluation and specific treatment. Acute tubular interstitial nephritis has been described in most cases of AKI in patients treated with immune checkpoint inhibitors and is treated with first-line corticosteroids. Lastly, BCG therapy can lead to disseminated BCGitis and acute interstitial nephritis, with or without granuloma, that requires antituberculous therapy and corticosteroids.

Readers must keep in mind that the future of anticancer immunotherapy remains to be written. New immune checkpoint targets and combinations [116,117], bifunctional checkpoint-inhibitory T cell engagers and trispecific killer engagers [94], CAR NK cell therapy [118,119] and new cancer vaccines [120] are likely to enter clinical practice in the decades to come along with yet embryonic strategies. These new treatments will also come with specific toxicities and undescribed renal side effects that nephrologists and oncologists will need to investigate and learn how to manage in order to provide the best standard of care for cancer patients.

## Figures and Tables

**Figure 1 cells-11-03991-f001:**
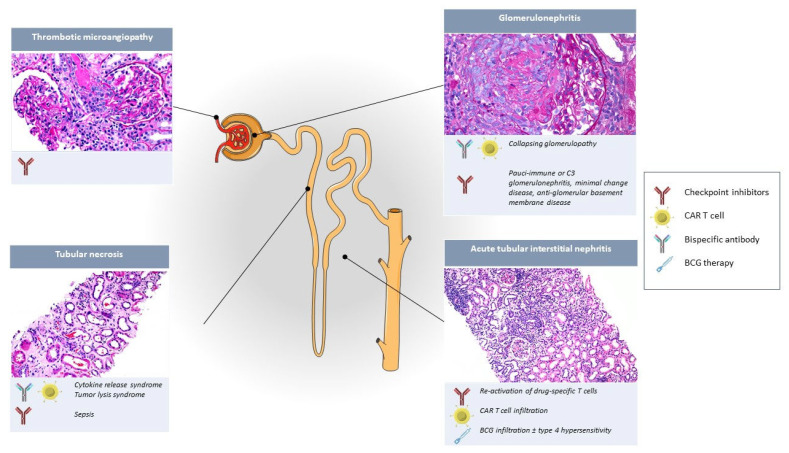
Acute kidney injury causes in cancer immunotherapy recipients and their site of injury in the nephron. We are grateful to Arkana Laboratories (Little Rock, AR, USA) for authorization to use renal pathology images.

**Table 1 cells-11-03991-t001:** List of approved cancer immunotherapies and their indications.

Immunotherapy	Class	Commercial Name	Indications
CAR T cells	Anti-CD19	Tisagenlecleucel (Kymriah^®^)	Acute lymphoblastic leukemia Diffuse large B cell lymphoma
		Axicabtagene ciloleucel (Yescarta^®^)	
		Lisocabtagene maraleucel (Breyanzi^®^)	
	Anti-BCMA	Idecabtagene vicleucel (Abecma^®^)	Multiple myeloma
Immune checkpoint inhibitors	Anti-PD1	Nivolumab (Opdivo^®^) Pembrolizumab (Keytruda^®^) Cemiplimab (Libtayo^®^)	Squamous head and neck cancer, lung cancer, melanoma, renal cell carcinoma, urothelial cancer, Hodgkin lymphoma and others.
	Anti-PDL1	Atezolizumab (Tecentriq^®^) Avelumab (Bavencio^®^) Durvalumab (Imfinzi^®^)	
	Anti-CTLA4	Ipilimumab (Yervoy^®^)	Melanoma, renal cell carcinoma
Bispecific antibodies	Anti-CD3/Anti-CD19	Blinatumomab (Blincynto^®^)	Acute lymphoblastic leukemia
	Anti-CD3/Anti-CD20	Mosunetuzumab (Lunsumio^®^)	Follicular lymphoma
BCG therapy	-	BCG (TheraCys^®^ and TICE^®^)	Non-muscle invasive bladder cancer

CAR: Chimeric antigen receptor, BCMA: B cell maturation antigen, BCG: Bacillus Calmette–Guérin.

**Table 2 cells-11-03991-t002:** Acute kidney injury associated with approved cancer immunotherapy.

Immunotherapy	AKI Incidence	AKI Mechanisms	Treatment
CAR T cells	18.6%	Cytokine release syndrome	Anti-IL-6 (siltuximab), IL-6R (tocilizumab) and corticosteroids
CAR T cell infiltration	
-

Tumor lysis syndrome	IV fluids, allopurinol, rasburicase
Immune checkpoint inhibitors	1–5%	Acute tubular interstitial nephritis	Corticosteroids±Second-line immunosuppressants
Acute tubular injury
Glomerulonephritis/Minimal change disease
Bispecific antibodies	1%	Cytokine release syndrome	Anti-IL-6 (siltuximab), IL-6R (tocilizumab) and corticosteroids
Tumor lysis syndrome	IV fluids, allopurinol, rasburicase
BCG therapy	2–3%	Acute interstitial nephritis ± granuloma	Anti-tuberculous tritherapy + corticosteroids

AKI: acute kidney injury, CAR: chimeric antigen receptor.

**Table 3 cells-11-03991-t003:** American Society for Transplantation and Cellular Therapy consensus grading for cytokine release syndrome and treatment (adapted from [15,20]).

CRS Grade	Grade 1	Grade 2	Grade 3	Grade 4
Temperature	Temperature > 38°			
*Treatment*	*Symptomatic measures, assessment for infection*			
**Hypotension**	None	Not requiring vasopressors	Requiring a vasopressor with or without vasopressin	Requiring multiple vasopressors (excluding vasopressin)
***Treatment* ***	*Hydration fluid*	*IV fluid bolus* *Tocilizumab 8 mg/kg, repeated after 6 h if necessary* *Dexamethasone 10 mg/6 h if hypotension persists after anti-IL-6 therapy*	*+* *Vasopressor as needed* *If refractory, increase dexamethasone 20 mg/6 h*	*+* *Methylprednisolone 1 g/day*
**Hypoxia**	None	Requiring low-flow nasal cannula or blow-by	Requiring high flow nasal cannula, facemask, nonbreather mask or Venturi mask	Requiring positive pressure (e.g., CPAP, BiPAP, intubation and mechanical ventilation

* Consider admission to an intensive care unit for CRS grade 2 or higher.

**Table 4 cells-11-03991-t004:** American society of clinical oncology grading and treatment of renal immune-related adverse events (adapted from [66]).

	Grade 1	Grade 2	Grade 3	Grade 4
Diagnosis	Creatinine level increase of > 0.3 mg/dL.; creatinine 1.5–2> above baseline	Creatinine 2–3× above baseline	Creatinine > 3 × baseline or > 4.0 mg/dL.; hospitalization indicated	Life-threatening consequences; dialysis indicated; creatinine 6× above baseline
Management *	Consider temporarily holding ICI	Hold ICI temporarily. Administer corticosteroids (0.5–1 mg/kg/day prednisone equivalents). If worsening or no improvement after 1 week, increase to 1–2 mg/kg/day and permanently discontinue ICI.	Permanently discontinue ICI Administer corticosteroids (initial dose of 1–2 mg/kg/d prednisone or equivalent). If elevations persist or worsen, consider additional immunosuppression (e.g., infliximab, azathioprine, cyclophosphamide (monthly), cyclosporine, and mycophenolate).	

***** Non-specific management include exclusion of potential alternative etiologies, fluid status optimization and nephroprotective therapy.

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
