# Peer review of "Acute Kidney Injury in Cancer Immunotherapy Recipients"

_cells, 2022, doi:10.3390/cells11243991_

Round 1

Reviewer 1 Report

-          General comments
Joseph et al. have focused on the description of several cancer immunotherapeutic strategies and how they can affect the kidneys causing acute kidney injury with different mechanisms.
The paper is clearly written, complete, and all topics are listed properly.

However, in some aspects (as described below) the review should have been more elaborated in order to give more information circa the clinical management of these peculiar AKI conditions.

Therefore we only suggest minor revisions, as listed below.

-          Specific comments

Abstract

- (line 9): two opportunities instead of “are being described”: “have been described” or “and nowadays renal side effects are described”.
- (line 11): we suggest to use “as a consequence” instead of “through the consequences”
- (line 14): “reproduce the spectrum of autoimmune kidney diseases” is too generic.

1. Introduction:

- (line 22) we suggest “the immuno-surveillance against cancer” instead of “the potential of the immune system in the fight against cancer”.
- (line 35) we suggest “..revolutionized the field of oncology becoming the first line of treatment..”

- (line 36) we suggest “.. wide range of indications. Immunotherapy is able to unleash..”

- (line 37-38) we suggest to remove “one patient’s” (immune system) and “one’s” (immune cells)
- Table 2 should not be inserted in the chapter “Acute kidney injury in CAR T cell recipients”, indeed Table 2 is a list of the approved cancer immunotherapies and their indications, so it would be more suitable in the “Introduction”.
- Table 2: L of lung should not be in capital letter.

- For three times the term “immune checkpoint blockers” is used, but it would be more uniform to use always “immune checkpoint inhibitors”.

- Table 1: Minimal change disease is a glomerulonephritis so it should not be listed separately

2. Acute kidney injury in CAR T cell recipients

-(line 58-59): we suggest “This unique architecture is able to recognize”

- (line 74): “..specific toxicities: cyotkine release syndrome..”

2.1 Cytokine release syndrome:

- In the heading of the chapter, the acronym “CRS” should be added, as it is used in the rest of the section.
- (line 97) It is reported that high grade CRS is associated with a higher risk of AKI; however, the criteria used in the quoted study to classify the severity of CRS are not reported and should be added.
- CRS treatment should be reported more in detail, in particular regarding indication, posology and duration of therapy with IL 6R and corticosteroids; even if there are no validated schemes, it would be enough to report the scheme purposed in the quoted study.

- (line 104): “..which, in this context, may have..” 

2.2 Tumor Lysis Syndrome:

- (line 127) The electrolytes disorders related to CAR T therapy and not secondary to tumor lysis syndrome (TLS), even if their etiology is not well defined and multifactorial, should be described. 

2.4Sepsis:
-(line 143) The incidence of fungal and viral infection is not reported.

-(line 148): It is not discussed the need for prophylactic therapy.

3 Acute kidney injury in immune checkpoint inhibitors recipients

- A Table of the immune-related adverse events (irAEs) with the severity criteria should be added.
3.1 Acute Interstitial Nephritis:

- This topic should be written using a different order of the sub-topics: (e.g) pathophisiology, risk factors (including concomitant drugs), clinical aspects, contraindications to ICI therapy, histopathological features, the controversial indications to kidney biopsy, the lack of association between histological features and kidney outcomes, treatment, readministration after AIN, clinical course.

- (line 190) Among the clinical aspect it should be highlighted the presence of non-selective proteinuria since it is not a glomerular process.

- (lane 206) The positivity of positron-emission tomography imaging for more than 2 weeks as a clinical indication to fore go kidney biopsy and start immunosuppression is questionable, in fact it might be questioned that this approach leads to delayed steroid treatment initiation which can lead to minor kidney function improvement.

- (line 225) the reported Odds Ratio refers to which risk factors?

- (line 231) It is not clear if Fluindione is associated to AIN with or without the concomitant therapy with ICI.

- (line 234) the content is not clear: it seems that in case of any immune-related nephropathy ICI therapy is contraindicated.

- (line 245): a table showing the nephrotoxicity severity criteria should be added.

- (line 258-259): the recovery rate is incomplete: indeed it is not reported how many patients require KRT, and in general (patient with or without the necessity to start KRT) how many patients develop  CKD.

- (line 273-274): since the available data are weak and since the outcomes are poorer if AKI requiring KRT develops, the message that the onset of AIN associated to ICI could be an indicator of ICI therapy effectiveness is not necessary and could be misleading.

3.2 Other:

- (line 280): we suggest to use “pauci-immune glomerulonephritis ANCA +/-“.

- (line 282): we suggest “.. membrane disease and C3 glomerulonephritis..”

Reviewer 2 Report

The article is comprehensive and well-written. The conclusion is not appropriate and it looks like the future treatment. the conclusion should reflect the summary of the article in my opinion and should be re-written.
